# Advances in Oral Subunit Vaccine Design

**DOI:** 10.3390/vaccines9010001

**Published:** 2020-12-22

**Authors:** Hans Van der Weken, Eric Cox, Bert Devriendt

**Affiliations:** Department of Virology, Parasitology and Immunology, Ghent University, Salisburylaan 133, 9820 Merelbeke, Belgium; hans.vanderweken@ugent.be (H.V.d.W.); eric.cox@ugent.be (E.C.)

**Keywords:** oral vaccination, subunit vaccines, mucosal immunity, secretory IgA, adjuvants, targeted delivery, Micro- and nanoparticles

## Abstract

Many pathogens invade the host at the intestinal surface. To protect against these enteropathogens, the induction of intestinal secretory IgA (SIgA) responses is paramount. While systemic vaccination provides strong systemic immune responses, oral vaccination is the most efficient way to trigger protective SIgA responses. However, the development of oral vaccines, especially oral subunit vaccines, is challenging due to mechanisms inherent to the gut. Oral vaccines need to survive the harsh environment in the gastrointestinal tract, characterized by low pH and intestinal proteases and need to reach the gut-associated lymphoid tissues, which are protected by chemical and physical barriers that prevent efficient uptake. Furthermore, they need to surmount default tolerogenic responses present in the gut, resulting in suppression of immunity or tolerance. Several strategies have been developed to tackle these hurdles, such as delivery systems that protect vaccine antigens from degradation, strong mucosal adjuvants that induce robust immune responses and targeting approaches that aim to selectively deliver vaccine antigens towards specific immune cell populations. In this review, we discuss recent advances in oral vaccine design to enable the induction of robust gut immunity and highlight that the development of next generation oral subunit vaccines will require approaches that combines these solutions.

## 1. Introduction

Vaccines play a crucial role in reducing the global burden of infectious diseases and are responsible for the elimination of diseases like polio, tetanus and pertussis and even the eradication of smallpox and rinderpest [1,2,3,4]. Furthermore, vaccines can aid in solving the current crisis regarding antimicrobial resistance by eliminating or reducing the need for antibiotics, especially in animal husbandry [5,6]. Most vaccines are administered via parenteral routes, generally leading to strong systemic immune responses. In contrast, most pathogens infect or invade the host at mucosal surfaces and systemic immunity generally does not provide sufficient protection against these types of pathogens. Local administration of vaccines to mucosal surfaces, such as via oral vaccination, provides much better protection against pathogens that colonize or invade these surfaces by inducing mucosal immunity, characterized by the local production of secretory IgA (SIgA) as well as a systemic immunity [7]. The production of SIgA is crucial because it shows improved stability in the gut via its secretory component and can prevent the colonization of the gut tissues by pathogens, such as enterotoxigenic *Escherichia coli* (ETEC), via a process called “immune exclusion”, characterized by agglutination, entrapment and clearance of the pathogen [8]. In addition to eliciting robust intestinal immune responses, oral vaccination has other advantages over parenteral vaccines, such as the reduced need for trained personnel, allowing self-administration, and a reduced risk of transmitting blood-borne diseases due to needle-free administration. They also increase patient compliance due to easier administration and often do not require refrigerated storage, which results in easier transport and delivery to remote places. For most oral vaccines, no expensive purification techniques or equipment are required, generally making it easier to get market approval. Finally, they also have a more cost-effective production, drastically reducing the cost of mass vaccination programs [9,10]. Currently, most oral vaccines consist of either inactivated or live-attenuated organisms. The latter have several risks attributed to them, such as uncontrolled replication, severe inflammatory reactions, the risk of infection in immunocompromised patients and the possibility of reversion to a virulent strain. In recent years, the focus for oral vaccination strategies has shifted to the use of safer subunit vaccines, but these still face many hurdles. Oral vaccine antigens have to survive the harsh environment of the gastrointestinal tract, characterized by a low gastric pH and degradation by gastric and small intestinal proteases. They also have to be able to reach the gut associated lymphoid tissue (GALT), which is protected by an epithelial barrier that has evolved to regulate nutrient absorption as well as to provide protection against foreign invaders [9]. Furthermore, under normal circumstances, antigens that enter via the oral route are treated as dietary components. If a vaccine does not induce the appropriate danger signals, then the gut tissues will recognize it as non-pathogenic, resulting in suppression of immunity or tolerance [11,12]. Compared to parenteral immunizations, high dosages are generally required for successful immunization, but these larger doses also increase the risk of tolerance [13,14,15]. Because of this risk, inclusion of potent adjuvants is essential for promoting robust intestinal immune responses [16]. The limited residence time of vaccine antigens in the gut is also an important factor to consider as it can prevent their effective uptake [17]. All these obstacles generally lead to poor immune responses to oral vaccination and are the main reasons so few effective oral vaccines exist. Finally, the microbiota might also impinge on the efficacy of oral vaccines. Currently, this aspect of oral vaccination is not yet well understood it certainly requires further research [18,19].

In this review, we provide an in-depth overview of the most important strategies that are being developed to tackle the many challenges associated with oral vaccination and give our opinion in what direction the design of oral subunit vaccines should evolve to unlock the full potential of oral vaccination.

## 2. Oral Vaccination Strategies

Several oral vaccination strategies have been developed in recent years to tackle the different hurdles associated with oral vaccination. Potent oral adjuvants have been developed that can stimulate the mucosal immune system and are capable of provoking robust mucosal immune responses. Different delivery systems have been designed that are able to protect vaccine antigens against the harsh gastrointestinal environment and release these antigens at the immune inductive sites to promote uptake by antigen presenting cells. Furthermore, by targeting specific receptors, selective delivery of vaccine antigens towards specific cell populations within the intestinal tissues can be achieved, further promoting robust intestinal immune responses. These oral vaccination strategies will be addressed in the next sections and are briefly summarized in Table 1.

### 2.1. Oral Adjuvants

As mentioned, most licensed oral vaccines make use of complete live-attenuated or inactivated organisms, which do not require potent adjuvants. These types of vaccines present an inherent adjuvanticity through the presence of conserved molecular patterns, such lipopolysaccharide (LPS), flagellin or cytosine-phosphate-guanine (CpG), which are recognized by pathogen recognition receptors (PRR) [80]. Since oral subunit vaccines generally do not possess any adjuvant functions, the addition of potent mucosal adjuvants is required to circumvent the default tolerogenic responses present in the gut and to allow the induction of robust intestinal immune responses [81,82]. Several types of adjuvants can be distinguished, and these can be broadly divided into two groups: the immunopotentiators and the delivery systems. Generally, immunopotentiators have the ability to enhance the immune response against otherwise weak immunogenic antigens and result in a broad and durable protection, while delivery systems improve the vaccine delivery to the targeted site or help protect the antigen from degradation. Often a combination of both these systems is used by including immunopotentiators in the delivery system or because the delivery system itself has inherent immune stimulating properties [83,84,85].

We will first focus on the different types of immunopotentiators, while the different types of delivery systems will be handled later in Section 2.

#### 2.1.1. Toxin Derivates

One of the most important classes of immunopotentiators for oral vaccination is toxin derivates, such as the ADP-ribosyl transferase enterotoxins cholera toxin (CT) and heat-labile enterotoxin (LT). Due to their strong adjuvant properties and their ability to elicit SIgA, they are considered the gold standard for oral vaccination [86,87]. These toxins stimulate antigen-presenting cells, enhancing the expression of MHC class II and costimulatory molecules, and induce antigen-specific T_H_2 and T_H_17 cells to secrete IgA-promoting cytokines, further supporting the production of IgA [88,89]. It has been shown that oral delivery of CT to mice activates the canonical NF-ĸB pathway and mRNA expression of NF-ĸB-dependent pro-inflammatory cytokines in the mesenteric lymph nodes and Peyer’s patches [90]. Their effects are also thought to promote the permeation of antigens across the epithelial barrier and to promote intestinal stem cells to differentiate into M cells, an epithelial cell specialized in the uptake of macromolecules [91,92,93].

Although LT and CT are often used as potent oral adjuvants in animal models during preclinical research, unfortunately they display a high toxicity in humans, resulting in severe diarrhea at low doses and thus preventing their use as oral adjuvants in humans. Fortunately, modified versions of these toxins, such as the double mutant LT (dmLT) and multiple-mutated CT (mmCT), have been developed in recent years, resulting in a decreased toxicity, while retaining their potent adjuvant properties [20,24]. In mice, dmLT has been shown to be an effective mucosal adjuvant when given orally together with several antigens from different pathogens, often providing protection against subsequent challenge infection [20]. In humans, a live-attenuated ETEC vaccine (ACE527) was co-administered with dmLT, resulting in protection after subsequent ETEC challenge [21]. The oral inactivated ETEC vaccine ETVAX also showed increased immune responses in children and infants, but not in adults when adjuvanted with dmLT [22,23].

An alternative strategy to co-administration would be to conjugate or fuse these toxins to non-immunogenic antigens, allowing the binding of the B-subunit to intestinal epithelial cells, resulting in uptake and transport through the epithelium and improved immunogenicity. Examples of this strategy in the literature include fusion proteins, such as fimbriae-toxin multi-epitope fusion antigens (MEFA) [94,95,96,97,98,99]. Although these showed promising results in inducing protective immunity after i.m and s.c. administration in mice, the protective efficacy of this vaccination strategy still needs to be assessed after oral administration and challenge infection.

Besides LT and CT, the potential use of other toxin derivates, such as adenylate cyclase toxins, as an adjuvant for oral vaccination still has to be further investigated. In mice, nasal co-delivery of the anthrax edema toxin with ovalbumin resulted in high antigen-specific IgG and IgA serum responses and induced antigen-specific T-cell secretion of IFNγ, IL-5, IL-6 and IL-13 [100].

#### 2.1.2. PRR Ligands

Pathogen recognition receptors play a crucial role in the recognition of pathogens and the induction of appropriate immune responses. They are expressed by many cell types, including intestinal epithelial cells and antigen presenting cells, such as dendritic cells and macrophages. In mice, the expression of some Toll-like receptors (TLR) by intestinal epithelial cells seemed to be age-dependent and differed along the length of the intestine, with the expression of TLR5 being restricted to Paneth cells in the small intestine and gradually decreased during the neonatal period [101]. PRR ligands have been intensively investigated for their adjuvanticity and can be subdivided in the membrane-bound TLRs and C-type lectin (CLR) receptors and the cytoplasmic RIG-I-like (RLR) and NOD-like (NLR) receptors (Table 2).

The ligands of CLR, NLR and RLR have not been studied well as oral adjuvants. So far, only β-glucans have been shown to have immune stimulating properties after oral administration [25,26]. The potential adjuvanticity of several TLR-ligands, such as monophosphoryl lipid A (MPL; TLR4), flagellin (TLR5) and CpG (TLR9) has been better studied [113,114]. MPL is a detoxified derivative of LPS [114]. Its interactions with TLR4 triggers the production of TNFα, IL-12 and IFNγ, promoting T_H_1 immune responses. In mice, pulmonary immunity against *M. tuberculosis* was obtained after oral administration of *M. tuberculosis*-derived antigens with MPL-based adjuvants [27]. The TLR5 ligand flagellin is a highly abundant protein in flagellated bacteria and promotes the induction of pro-inflammatory cytokines and chemokines, the recruitment of B- and T-cells to secondary lymphoid tissues, the direct activation of T-cells and the activation of DC’s [115,116]. Flagellin produced in plants was shown to be a potent adjuvant after oral administration with ovalbumin in mice [28]. Flagellin-coated ovalbumin-containing nanoparticles were found to enhance SIgA antibody responses to ovalbumin after oral administration in mice [29]. In humans, an influenza-flagellin fusion vaccine (VAX125) provided strong systemic immune responses after intramuscular immunization [117,118]. CpG is a synthetic oligodeoxynucleotide composed of unmethylated CG motifs. Its binding to TLR9 triggers the secretion of pro-inflammatory and T_H_1-specific cytokines by DC’s, facilitating the induction of cell-mediated immunity. CpG also promotes the maturation and proliferation of NK cells, T-cells and monocytes/macrophages [119,120]. In mice, oral administration of purified hepatitis B surface antigen or tetanus toxin adjuvanted with CpG provided both systemic and mucosal immune responses [30]. In piglets, oral vaccination with a live-attenuated pseudorabies virus, adjuvanted with CpG, resulted in significantly higher serum IgG and mucosal IgA responses compared to piglets that did not receive the adjuvant [31]. Antigen-presenting cells have also been found to make a distinction between living or dead cells via TLR8-dependent detection of bacterial RNA, resulting in the differentiation of follicular T-helper cells. TLR8-agonists, such as CL075 or R848 showed similar responses and might hold promise as oral adjuvants [121].

An important observation that argues against the use of PRR ligands for oral vaccination is that the intestine already continuously encounters these ligands, which could result in hypo-responsiveness or the presence of a higher threshold for PRR ligand-mediated cellular activation.

#### 2.1.3. Other Immune Modulating Molecules

Other immune modulating molecules that have been investigated as adjuvants include Natural Killer T (NKT) ligands and stimulator of interferon genes (STING) ligands. NKT ligands, such as the synthetic α-galactosyl ceramide, activate NKT-cells by binding to the CD1d receptor on antigen presenting cells. This α-galactosyl ceramide-CD1d complex is subsequently recognized by the NKT T-cell receptor. Mucosal tissues contain many NKT-cells that secrete both T_H_1, T_H_2 and T_H_17-specific cytokines upon stimulation. Alpha-galactosyl ceramide has been shown to be an effective adjuvant, inducing mucosal and systemic cell-mediated immunity after nasal or oral delivery with HIV peptide antigens in mice [32,122]. Addition of α-galactosyl ceramide to the oral cholera vaccine Dukoral^®^ also strongly enhanced intestinal immune responses in mice [33].

STING ligands, such as cyclic dinucleotides of bacterial origin (2′,3′-cGAMP, 3′,3′-cGAMP, c-di-AMP and c-di-GMP), can stimulate robust type 1 interferon responses and proinflammatory cytokines, such as TNFα, IL-1β and IL-6, resulting in the activation of macrophages and dendritic cells. These primarily showed promising results with intranasal use and it would be interesting to assess their efficacy in oral vaccination [123,124].

#### 2.1.4. Use of Adjuvants for the Induction of SIgA after Parenteral Administration

Two factors are crucial for inducing the production of SIgA at the induction sites. First, cytokines play an important role in driving the differentiation of T-helper cell populations, permitting intestinal immunity. Secondly, gut homing of effector cells, like plasma cells, towards the mucosal effector sites is another crucial step.

Gut homing is orchestrated by the expression of mucosal addressins, integrins, chemokine receptors and their ligands [125,126]. They allow the migration of activated lymphocytes and antibody-secreting cells towards specific regions in the gut. Both the integrin α4β7 and the chemokine receptor CCR9 are known to regulate gut homing of immune cells towards mucosal tissues in the gut [127,128]. The integrin α4β7 is present on activated T and B cells and allows binding to the mucosal vascular addressin cell adhesion molecule-1 (MAdCAM-1), expressed on endothelial cells in the high endothelial venules (HEV) of the small intestine and Peyer’s patches [129]. The chemokine receptor CCR9 is also present on T-cells and binds specifically to the chemokine CCL25, expressed within the crypts and lower villi of the small intestinal epithelium and on the surface of vascular endothelial cells in the small intestine [130,131,132]. An important molecule involved in gut homing is all-trans retinoic acid (ATRA). This vitamin A metabolite is primarily produced by CD103^+^ DCs and enables these cells to imprint the expression of α4β7 and CCR9 on lymphocytes. In the absence of ATRA, differentiation of IgA-producing cells will lead to the induction of α4β1, L-selectin and CCR10, which targets B-cells to other mucosal tissues, like the airways, salivary glands, reproductive organs or the colon [133,134].

Successful oral immunization should result in the activation of intestinal dendritic cells that produce high amounts of ATRA, leading to the generation of IgA-secreting cells capable of migrating towards the intestinal mucosa. Several factors influence the immune stimulating effects of ATRA, including IL-5, IL-6 and IL-21 or sphingosine 1-phosphate. Both IL-5 and IL-6 synergistically modulate the IgA producing effects of ATRA by modulating IgA class switching in a T-cell independent manner [135,136,137]. IL-21 production can be triggered by IL-6 and drives plasma cell differentiation. Expression of sphingosine 1-phosphate is regulated by ATRA signaling and is needed for the egression of immune cells from the lymphoid organs into the lymphatic vessels [138,139,140,141,142,143]. Besides gut homing and antibody-class switching, ATRA also promotes the generation of regulatory T cells and inhibits the differentiation of T_H_17 cells by enhancing TGFβ signaling [144,145].

Although ATRA is not necessarily considered as a mucosal adjuvant, its function could be important for the development of vaccines aiming at eliciting robust mucosal immune responses. Upon subcutaneous or intraperitoneal administration of ATRA together with vaccine antigens, increased α4β7 and CCR9 expression on lymphocytes and increased T-cell trafficking towards the gut were observed in mice and pigs [146,147,148]. More recently, a vaccine using two liposomal delivery systems that were subcutaneously administered to mice induced antigen-specific intestinal IgA responses. The first delivery system was designed to ensure fast drainage of ATRA towards the lymph nodes to precondition these for mucosal immune responses, while the second delivery system was optimized for slower, prolonged delivery of the antigen to these ATRA preconditioned lymph nodes via migrating antigen-presenting cells [149]. In the future, it would be interesting to see if similar results could be obtained in large animal models.

### 2.2. Delivery Systems

Due to the harsh environment in the gastrointestinal tract, antigens need to be protected from the low pH and degradation by proteases. Several types of delivery vehicles and encapsulation strategies have been developed to protect and preserve the structural integrity of antigens and enable their release within the inductive sites to facilitate phagocytosis by antigen presenting cells. Besides protecting and delivering the antigens, they also often contain adjuvants that can enhance the immune response and prevent tolerance [85]. In general, these delivery systems can be divided in living and non-living systems.

#### 2.2.1. Living Delivery Systems

Living delivery vehicles mostly include live bacterial strains and viral vectors that do not infect the host. They generally have strong adjuvant properties based on the recognition of their PAMPs by PRRs. This in turn results in the release of inflammatory cytokines, inducing the production of SIgA in the intestinal tissues.

##### Recombinant Bacteria

Recombinant bacterial strains used as antigen delivery vehicles in oral vaccination include lactic acid bacteria, *Salmonella* and *L. monocytogenes* strains [150]. Several recombinant strains have been developed that express one or more antigens derived from a variety of pathogens to induce protection against the corresponding pathogens. Examples include vaccines against tetanus, *H. pylori*, enterohemorrhagic and enterotoxigenic *E. coli* (EHEC/ETEC), *C. difficile*, *S. enterica*, rotavirus, *C. albicans*, avian influenza virus and even parasites, such as *Plasmodium yoelii* and *Giardia lamblia* [34,35,36,37,38,39,40,41,42,43,44,151,152,153]. A disadvantage is that these recombinant bacterial strains must be engineered to only survive inside the host, since they are genetically modified organisms (GMO). A major advantage of using lactic acid bacteria is that these strains are generally regarded as safe, allowing their oral consumption.

##### Viral Vectors

Besides bacteria, several viral vectors have been developed to aid in antigen delivery. By genetically modifying them to express foreign antigens, viral vectors can be used as vaccine delivery vehicles to promote antigen-specific immune responses. Examples include the pox and measles viruses, alphaviruses, baculoviruses, adenoviruses or adeno-associated viruses [45,46,154,155]. They generally promote immune responses via stimulation of PRRs, such as TLR3, 7 and 8 or the RLRs, RIG-I and MDA-5 [156]. The most promising viral vectors for oral vaccination in humans are recombinant adenoviral vectors [157,158,159,160,161]. Adenoviruses are non-enveloped, double stranded DNA viruses that are capable of eliciting strong T -and B-cell responses. Most adenoviruses cause mild disease symptoms in immunocompetent humans and are generally safe to use. They can also replicate in virtually all living cells, including cell lines, making them easy to produce. Furthermore, they have a packaging capacity of up to 35 kb, allowing large inserts [162]. To reduce unwanted side-effects, replication-deficient vectors have been designed by deleting replication-specific regions of the adenoviral genome [163]. Oral vaccination with human adenovirus 4 and 7 has been used to confer protection against adenoviral respiratory tract infection [164]. Adenoviral vectors under investigation for human vaccination are mostly derived from the human serotype 5, but also from the human serotype 26, simian serotypes 23 and 24, or the chimpanzee serotypes 6 and 7 [165]. A concern associated with the use of human adenoviral vectors is the existence of pre-existing immunity against these vectors in humans, which might reduce their efficacy [163]. Several oral adenoviral-based vaccines are currently under investigation and undergoing clinical trials against diseases such as HIV, influenza, respiratory syncytial virus, norovirus and the human papilloma virus (HPV) [47,48,49,50,51,52,53,54,166,167]. Currently, an oral adenoviral-based vaccine targeting the severe acute respiratory coronavirus 2 is also in development at a preclinical stage [168]. In veterinary medicine, this technology has advanced further, with several effective vaccines on the market against diseases such as rabies or the pseudorabies virus in pigs [55,56,169].

#### 2.2.2. Non-Living Delivery Systems

In addition to living delivery vehicles, several non-living delivery vehicles have also been developed. These generally include virus-like particles, micro/nanoparticles and nanogels. These systems are generally less potent than living systems in inducing immunity, since they lack potent PRR ligands and are unable to replicate.

##### Virus-Like Particles

Virus-like particles (VLP) are molecules that resemble viruses but are not infectious because they lack genetic material. They are able to self-assemble after synthesis of their viral structural proteins and combinations of different viruses can be used to create recombinant VLPs [170,171]. Many heterologous antigens can be placed on their surface, allowing immune stimulation. The earliest examples of VLPs include vaccines against hepatitis B and HPV-induced cervical cancer via the hepatitis B virus surface antigen and the HPV capsid protein L1, respectively [172,173]. It was shown that VLPs fused with variant-specific surface proteins (VSP) from species such as *Giardia lamblia* were protected from extreme pH, temperatures and proteolytic digestion after oral administration. Furthermore, they stimulated host innate immune responses in a TLR-4 dependent manner, based on the CXXC-motif of VSPs [57].

##### Micro- and Nanoparticles

Polymeric micro-/nanoparticles and lipid-based vehicles can also aid in delivering vaccine antigens to the induction sites. They are able to protect their delivered antigens from degradation and show a strong uptake and internalization by antigen-presenting cells and subsequent immune stimulation. By manipulating their surface properties, these delivery systems allow the targeted delivery to specific immune populations. Furthermore, the adjuvant functions of other immunomodulators can be enhanced. Both synthetic and natural materials exist with different physicochemical properties, providing versatile options for oral vaccine design [58,59]. Natural materials include chitosan, starch, alginate, cellulose, β-glucan yeast ghost particles or biosynthesized poly β-hydroxybutyrate. Synthetic materials include polyurethane, polylactic acid, poly (lactic-co-glycolic acid) (PLGA) and polymethyl methacrylate resins [60,174,175,176,177,178]. Especially natural materials have been extensively investigated for oral vaccine delivery due to their high biodegradability and compatibility, low to non-existing toxicity, strong adjuvanticity effects and their ability to retain the conformational structure of their loaded antigen [59]. Cellulose acetate phthalate (Eudragit) for example is often used as a polymer film to protect a capsulated vaccine as it is insoluble at the low pH of the stomach, but dissolves readily at higher pH in the gut, depending on the type of Eudragit [179].

##### Lipid-Based Delivery Systems

Lipid-based vehicles are spherical vehicles, composed of at least one phospholipid bilayer and include liposomes, bilosomes and immune stimulating complexes (ISCOM). Liposomes can be prepared in several ways, allowing protein antigens or nucleic acids to be loaded into the vehicles [180]. After delivery, their lipid bilayers can fuse with other bilayers, such as cell membranes, resulting in the delivery of their contents [181]. They can be optimized by modulating membrane compositions using neutral, cationic or anionic lipids. This allows high stability, controlled release, low toxicity, improved adjuvant properties or longer blood circulation half-life [182]. Several liposome-based vaccines have been developed in recent years, against pathogens such as influenza A, *M. tuberculosis*, *S. Enteritidis* and group A Streptococcus [61,62,63,64].

Bilosomes are liposomes that were specifically designed for oral vaccination. Their phospholipid bilayer contains bile salts, like sodium deoxycholic acid, that stabilizes and protects the bilosomes and its contents from premature release in the hostile gastrointestinal tract [183,184]. Examples of bilosome-based formulations include oral vaccines against hepatitis B, cholera, tetanus, Influenza A and the human enterovirus 71 [45,65,66,67,68,69].

ISCOMs are a variant of liposomes in which cage-like structures are formed made of cholesterol, saponins and phospholipids. These can then in turn entrap the antigens and protect them from degradation. It should be noted however that ISCOMs are not ideal for the oral route, since the vesicles are labile and readily disassembled by detergents in the gut. Additionally, because of their small internal size, antigens are usually incorporated into the outer membrane, restricting its use to membrane-bound proteins [185,186].

##### Nanogels

Finally, nanogels are composed of a crosslinked hydrophilic polymer network or hydrogel. The pores in these nanogels can be filled with antigens, which in turn are protected from low pH. They can be designed to release the antigen when pH values rise in the intestine, resulting in efficient delivery to the induction sites [187]. Although some nanogel-based vaccines have reached clinical phase trials in human studies, they have only been assessed as an oral vaccine delivery system in mice [70].

### 2.3. Antigen Delivery to the Intestinal Immune System

After antigens have survived the harsh gastrointestinal environment, they need to be taken up by antigen-presenting cells to be presented to effector immune cells. Due to its close contact with the outside world, the intestinal epithelium has developed several physical and chemical barriers that are pivotal for immune homeostasis, but make it difficult for oral vaccines to reach their intended targets and induce robust intestinal immune responses. To surmount these problems, selective delivery of oral vaccines to specific cell populations in the intestinal mucosa is being investigated. Approaches for improving oral vaccine delivery in this manner include targeting of vaccine antigens towards receptors on M cells, intestinal epithelial cells and antigen-presenting cells using a variety of ligands.

#### 2.3.1. Microfold Cells

M cells are intestinal epithelial cells, present in the follicle-associated epithelium overlaying the Peyer’s patches, specialized in the uptake and transport of macromolecules and particulate matter from the small intestinal lumen to the gut-associated lymphoid tissue. As such, they represent an interesting target for vaccine delivery [178]. Several M cell markers have been studied to promote uptake of oral vaccines [188,189]. Interesting targets specifically expressed on M-cells include sialyl Lewis A-containing carbohydrates, dectin-1, GP2 and the complement 5a receptor (C5aR) [71,190,191,192]. The sialyl Lewis A is present on human M cells. Although this could be an interesting target for human vaccine development, to our knowledge, no sialyl Lewis A-mediated vaccine delivery systems have been designed that could direct antigens to M cells. As a CLR, dectin-1 recognizes carbohydrates, such as β-glucans and a specific glycosylation moiety on SIgA. After oral administration, SIgA complexed with the HIV p24 antigen was able to stimulate mucosal and systemic antibody responses in a dectin-1-dependent manner in mice [71]. GP2 is expressed on mouse and human M cells and can specifically recognize FimH, a component of type I pili on the bacterial outer membrane of specific enterobacteria. Uptake of FimH+ bacteria by M-cells via GP2 was able to initiate mucosal immune responses in mice [193]. Likewise, vaccines based on FimH were able to enhance humoral immune responses and block FimH-dependent bacterial adhesion in mice and cynomolgus monkeys [72]. To our knowledge, no studies have been performed that direct antigens to M cells in a FimH-dependent manner. The bacterial outer membrane protein H (OmpH) is a ligand of C5aR on human and mouse M cells. Oral administration of the EDIII antigen from dengue virus, conjugated with the OmpH ligand, resulted in EDIII-specific systemic and mucosal immune responses in mice [73]. Despite these advantages of M cells, these cells represent only a minor cell population in the gut epithelium and are located at the distal end of the small intestine.

#### 2.3.2. Enterocytes

In addition to M cells, enterocytes can also be targeted, as these cells are by far the most abundant cell type in the intestinal epithelium and are able to transcytose macromolecules, such as cholera toxin, F4 fimbriae, immune complexes and even inert particles [75,194,195]. Enterocytes are also able to phagocytose bacteria across the epithelial barrier in a TLR4-mediated manner and deliver them towards underlying antigen-presenting cells [196]. In addition, enterocytes express the neonatal Fc receptor (FcRn), which mediates the transcytosis of IgG-antigen immune complexes across the epithelial membrane due to its binding of IgG at acidic pH (pH < 6.5), enabling transport to the basolateral side [197,198]. Orally administered Fc-conjugated nanoparticles targeted to FcRn have been efficiently transcytosed across the epithelial barrier and could reach systemic circulation [74,199].

Another interesting receptor is aminopeptidase N (APN/CD13). This homodimeric transmembrane protein is highly expressed on enterocytes, where it represents around 8% of the total membrane proteins within the intestinal brush border membrane. Here, it is involved in digestive processes by removing N-terminal amino acids from peptides [200]. APN is also expressed on specific subsets of dendritic cells in humans, pigs and mice, which play a central role in the induction of adaptive immune responses [201,202,203]. APN serves as an important receptor for several pathogens via its extracellular regions, resulting in their attachment or internalization. These include coronaviruses from several species, the human cytomegalovirus (HCMV) and the bacteria *M. tuberculosis* and F4^+^ ETEC [76,204,205,206,207,208]. What makes this receptor so interesting for targeted vaccine delivery is that upon binding of F4 fimbriae to APN, these fimbriae were transported across the epithelium and induced strong immune responses after oral administration [75,76,77]. Moreover, when these fimbriae were used as a carrier system for conjugated human serum albumin, strong mucosal and systemic immune responses were obtained against this conjugated antigen [78]. Unfortunately, these F4 fimbriae cannot be used as a universal carrier system, since the binding to APN depends on the presence of specific carbohydrates, which are absent in certain piglets and other species [76,209,210]. However, antibody-mediated targeting to porcine APN, independent of this carbohydrate moiety, also triggered epithelial transcytosis. Oral administration to piglets of different APN-specific antibody formats (polyclonal, monoclonal as well as VHH-based antibody constructs) elicited systemic and intestinal antibody responses, further validating APN as an interesting receptor for targeted delivery of vaccine antigens [76,79]. The antibody-mediated targeting of antigen-loaded microparticles towards APN also showed promising results, improving uptake and subsequent antigen-specific immune responses upon oral administration [60]. Together, these data demonstrate that APN is an interesting receptor for the targeted delivery of vaccine antigens and suggests that a universal vaccine carrier system could be developed by targeting oral subunit vaccines towards APN.

#### 2.3.3. Antigen-Presenting Cells

Besides the targeting of epithelial cells, the underlying antigen-presenting cells, vaccine antigens, can also be selectively delivered to dendritic cells. These antigen-presenting cells express a broad spectrum of cell surface receptors that are involved in endocytose and can trigger DC maturation. These could be used to selectively deliver antigens to DC’s and promote maturation. Examples include C-type lectins, TLRs and Fc-receptors. Within the CLR family, DEC205, the DC-specific intercellular adhesion molecule 3-grabbing nonintegrin (DC-SIGN), dectin-1, the mannose receptor or Clec9A have been investigated to target vaccine antigens to dendritic cells [211]. Several DEC205-targeted vaccine strategies have been developed, but none for oral administration so far [212,213,214]. Similar to DEC205, targeting of antigens towards dectin-1 and the mannose receptor resulted in antigen presentation, but required adjuvant co-administration. In the absence of adjuvants, targeting of antigens towards these receptors resulted in the generation of regulatory T-cells, resulting in tolerance [215,216,217]. Targeting towards Clec9A, a receptor involved in the recognition of dead cells, on the other hand resulted in strong antibody responses in the absence of adjuvants [218]. The intercellular DC-SIGN was proposed to be involved in immune surveillance as SIgA-antigen immune complexes could be actively transported by M cells from the lumen towards dendritic cells [219]. It has been used to target vaccine antigens and lentiviral vectors towards DCs in mice, improving immunogenicity [220,221]. So far, these strategies have not been used to target intestinal DCs.

Besides targeting DC-specific receptors, the targeting of antigens towards Fcγ receptors (FcγR) present on dendritic cells and many other immune cells has also been shown to enhance humoral and cellular immune responses. Targeting of activating FcγR’s (FcγRI, FcγRIIa, FcγRIII) by monoclonal antibodies conjugated with an antigen enabled Fc-domain-mediated uptake by FcγR-expressing cells, such as B-cells, dendritic cells, neutrophils, macrophages, NK-cells and mast cells [222,223,224]. Here, it is important to note that one should try to avoid targeting the inhibitory FcγRIIb, since this could potentially dampen the signal [225,226].

Another interesting Fc-receptor for the targeting of antigen-presenting cells is FcRn. In addition to its expression by intestinal epithelial cells, discussed previously, FcRn is also expressed by antigen-presenting cells in several species [227,228,229]. Targeting of Fc-coated microparticles or antibody–antigen conjugates could allow the transfer through the epithelial barrier and target vaccine antigens to FcRn or FcγR-expressing DC’s. The herpes simplex virus-2 glycoprotein gD, fused with an IgG Fc fragment in addition with the CpG adjuvant was able to provide protection in mice after intranasal immunization. This occurred in an FcRn-specific manner and provided both mucosal and systemic antibody responses [197].

### 2.4. The Effect of Microbiota and Other Factors on Oral Vaccination

Recent years have shown the importance of the gut microbiota in regulating many aspects of both the gut and systemic immune system. Despite this, the impact of the microbiota on the efficacy of oral vaccines is often neglected. Recent studies have shown that the gut microbiota can influence oral vaccine efficacy [19,230]. During early life, the intestinal microbiota shapes the immune system [18]. Moreover, metabolites produced by the microbiota contribute to the integrity of the epithelial barrier, while dysbiosis of the intestinal microbiome contributes to vaccine failure [19,231]. The microbiota constitutes as a constant source of adjuvants, such as flagellin, LPS or bacterial peptidoglycans. Therefore, the composition of the microbiome might play an important role in how the immune system is modulated and can thus impact the response to vaccination. In the elderly, the gut microbiota is often in a state of dysbiosis, resulting in impaired production of immune stimulating metabolites [232]. In humans, Actinobacteria have been linked with increased humoral and cellular responses to BCG, tetanus toxin, hepatitis B virus and polio virus vaccination in Bangladeshi children, while Enterobacter was linked with reduced immune responses [233]. The composition of the infant gut microbiome has been linked with immune responses to the rotavirus vaccine [234]. Oral antibiotics can alter vaccine-induced immune responses and antibiotic-driven bacterial imbalance can lead to impaired immune responses [235]. Due to the emerging importance of the microbiota, supplementation with probiotics has been investigated for improving vaccine efficacy. In a systematic review that compared several studies that investigated the effect of probiotic strains on the response of different vaccines, beneficial effects were reported in around half the cases. Here, evidence of beneficial effects for probiotics were strongest for parenteral influenza vaccination and for oral vaccines [236]. A large part of the variation observed in vaccine responsiveness within populations could be attributed to differences in microbiota composition, but the molecular mechanisms underlying these differences are currently not well understood. Further research in this domain is definitely required as it could aid in developing more efficacious oral subunit vaccines [230].

Besides the microbiota, sexual, racial and health factors have also been described as influencing the effectiveness of vaccination. Females often develop higher antibody titers and cell-mediated immune responses upon vaccination, but also develop more adverse effects [237,238,239,240]. Several immunological, genetic, hormonal and environmental factors have been described to impact on these differences [241,242]. Racial or ethnic differences have also been described to contribute to vaccine efficacy [243,244,245,246,247,248,249,250]. Race-related differences in antibody and B-cell responses to the inactivated influenza vaccine could be linked to specific gene expression profiles, such as a differential expression of the programmed cell death-1 and the B and T cell attenuator on B-cells [251]. Health factors play a crucial role in vaccine efficacy in developing countries. Several health factors, such as systemic inflammation, maternal health and environmental enteropathy, characterized by intestinal inflammation, reduced intestinal absorption and dysfunctions in the gut barrier have been found to negatively impact the efficacy of the oral polio and rotavirus vaccines in Bangladeshi children [252]. This metabolic dysfunction leads to dysbiosis in the gut microbiota, exacerbating the negative effect on oral vaccines [253]. Mycotoxins, toxic secondary metabolites from various molds that often contaminate feed have also been found to reduce the efficacy of oral vaccines in animal models [254,255]. These factors might be considered in the design of next generation oral vaccines.

## 3. Expert Opinion

Despite the immense promise that oral vaccination holds, there are still relatively few oral vaccines on the market. Current existing oral vaccines for humans mainly consist of live-attenuated or inactivated organisms. Living vectors possess some inherent problems, such as the risk of reversion to virulence or ethical concerns since these are often genetically modified organisms being released into the environment, while inactivated vaccines generally provide weaker immune responses and do not often provide long-term immunity.

Subunit vaccines offer an interesting alternative in this regard. Unfortunately, the development of oral subunit vaccines is impeded by challenges associated with this administration route, such as instability issues in the gastrointestinal tract, a poor crossing of the epithelial barrier and a poor induction of robust mucosal immune responses. Current efforts are mainly focused on using different encapsulation strategies to preserve antigen stability in the gut, novel mucosal adjuvants to prevent the induction of oral tolerance and the targeting of vaccine antigens to specific intestinal cell populations to enhance uptake. Although a lot of progress has been made in recent years, further research is still needed to unlock the full potential of oral subunit vaccination. Currently, veterinary vaccines are at a more advanced stage due to less strict regulations. In the coming years, human vaccines will surely follow this trend, with several adenoviral vector vaccines already in later stage clinical trials.

In the future, vaccination strategies that combine different techniques will be required to deal with the different challenges associated with oral vaccination. This combinatorial approach (Figure 1) entails the use of nano- or microparticles that protect vaccine antigens from the gastrointestinal environment and aid in their delivery to immune induction sites. The choice of particles would depend on the possibility to include potent adjuvants to surmount tolerogenic responses. A versatile system is needed that allows the incorporation of molecules with different physicochemical properties. These delivery systems could be functionalized with targeting ligands, such as antibody formats, to promote transcytosis across the epithelial barrier and at the same time enhance uptake by antigen-presenting cells and promote their maturation. The design of such a combination strategy might expedite the development and use of oral vaccines to promote animal and human health.

## Figures and Tables

**Figure 1 vaccines-09-00001-f001:**
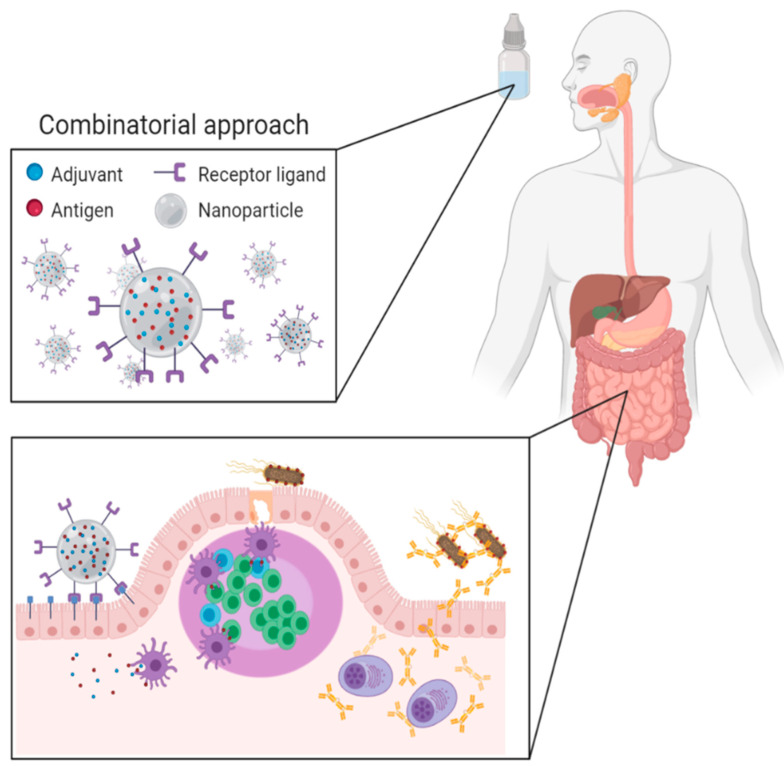
Combinatorial approach for novel oral vaccines: Antigen and adjuvant-loaded, receptor-targeted nanoparticles protect the vaccine antigens from the harsh intestinal environment and mediate the targeted delivery of antigens towards specific intestinal cell populations, thus promoting the induction of mucosal immune responses (Created with BioRender.com).

**Table 1 vaccines-09-00001-t001:** Overview of different oral vaccination strategies studied for oral administration.

**Oral adjuvants**	
Toxin derivates	dmLT [20,21,22,23], mmCT [24]
PRR ligands	β-glucans [25,26], MPL [27], Flagellin [28,29], CpG [30,31]
NKT-ligands	α-galactosyl ceramide [32,33]
**Delivery systems**	
Living delivery systems	Recombinant bacteria [34,35,36,37,38,39,40,41,42,43,44] Viral vectors [45,46,47,48,49,50,51,52,53,54,55,56]
Non-living delivery systems	Virus-like particles [57]
	Micro- and nanoparticles [58,59,60]
	Lipid-based delivery systems [61,62,63,64,65,66,67,68,69]
	Nanogels [70]
**Targeted delivery**	
M-cells	Dectin-1 [71], GP2 [72], C5aR [73]
Enterocytes	FcRn [74] Aminopeptidase N [60,75,76,77,78,79]

**Table 2 vaccines-09-00001-t002:** Overview of pathogen recognition receptors, with their respective cellular localization and ligands.

Pathogen Recognition Receptor (PRR)	Cellular Localization	Ligand
**Toll-like receptors (TLR)** [102,103,104,105,106]		
TLR1	Plasma membrane	Peptidoglycans/lipoproteins
TLR2	Plasma membrane	Peptidoglycans/lipoproteins
TLR3	Endosome	dsRNA
TLR4	Plasma membrane	LPS
TLR5	Plasma membrane	Flagellin
TLR6	Plasma membrane	Lipoproteins
TLR7	Endosome	ssRNA
TLR8	Endosome	ssRNA
TLR9	Endosome	Unmethylated CpG
TLR10	Endosome	Unknown
**NOD-like receptors (NLR)******* [107,108]		
NOD1/2	Cytoplasm	Peptidoglycans
NLRP3	Cytoplasm	PAMP, DAMP **
NLRC4	Cytoplasm	Cytosolic flagellin
**C-type lectin receptors (CLR)** [109]		
Dectin-1	Plasma membrane	β-glucans
Clec9A	Plasma membrane	F-actin
DC-SIGN	Plasma membrane	Mannose
Mannose receptor	Plasma membrane	Glycans
**RIG-I-like receptors (RLR)** [110]		
RIG-I	Cytoplasm	dsRNA
MDA-5	Cytoplasm	dsRNA

* Many more NLRs exist (NLRP1-14, NLRC1-5, NAIP, CIITA), but most of these have not been extensively researched [111]. ** Many different pathogen and damage associated molecular patterns are able to activate the NLRP3 inflammasome. This has been excellently reviewed by Kelley et al. [112].

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
