# Peer review of "Advances in Oral Subunit Vaccine Design"

_vaccines, 2020, doi:10.3390/vaccines9010001_

Round 1

Reviewer 1 Report

This review article by Van der Weken et al titled “ Advances in oral subunit vaccine design ” discussed recent advances in oral vaccine design and current challenges with oral vaccination regarding administration route and mucosal immune responses. Van der Weken et al provided up to date information known on several vaccination strategies using different oral adjuvants and different delivery systems. I strongly applaud authors work in gathering this information and hopefully the challenges with oral vaccination will be addressed.

Additional comments:

This review article by Van der Weken et al titled “ Advances in oral subunit vaccine design ” discussed recent advances in oral vaccine design and current challenges with oral vaccination regarding administration route and mucosal immune responses. As majority of infectious agent’s access host through the mucous membrane, mucosal vaccine delivery is important to develop local immunity as well as systemic immunity. Oral vaccines are easier to administer, cost-effective and are safe compared to the use of needles by avoiding injury and risk of infections.  Oral polio vaccine is one of the examples of oral vaccination against a highly contagious human enterovirus, which has success in almost all countries. Van der Weken et al addresses the importance of oral vaccines through this review.

Van der Weken et al put together up to date information known on several oral vaccination strategies using different oral adjuvants and different delivery systems supporting their conclusions. They addressed the importance of developing oral vaccines and current challenges associated with developing oral vaccines due to the obstacles faced while travelling through the GI tract. To address the harsh conditions in the GI tract,  Van der Weken et al mentioned the targeted delivery to the cells. This article is well written and easier to understand. I would like the authors to give a tabular presentation on different oral vaccine preparations and delivery preparations studied up to now. This will give reader’s clear understanding.  I strongly applaud authors work in gathering this information and hopefully the challenges with oral vaccination will be addressed.

Reviewer 2 Report

The review by van der Weken et al is an extensive overview of all aspects regarding oral vaccination. The review is well-written and contains all subjects related to this subject.

No further changes required.

Additional comments:

The main question discussed in the review is how protective SIgA responses can be evoked by oral vaccination and how the harsh environment in the gastrointestinal tract, characterized by low pH and intestinal proteases can be dealt with in order to reach the gut-associated lymphoid tissues. This is an important question in this time were there is a lot of discussion of vaccination and how the best vaccination strategies can be obtained.

The topic is original and approached from different angles. Next to describing delivery systems and vaccination strategies the authors also discuss the immunological part of vaccination and immune cells and systems involved. Especially that part is to my opinion the strongest part of the review.

I think the paper is well written and good to read. Of course there many technical terms are used but they are well explained. I could not detect flaws in the English that was used.

The authors acknowledge that much of the data is from animal studies and that there is still a long way to go, but that important progress has been made and that in the future oral vaccination may become more and more important.

Author Response

We thank the reviewer for these comments and appreciate that no further changes are required.

Reviewer 3 Report

General: This manuscript describes the development and immunogenicity of oral vaccines and vaccine components that can be used to tailor specific responses. The authors describe different strategies to induce immune responses including adjuvants, delivery systems, while also offering insight into the effects of the cells of the intestinal immune system including the microbiota. Overall this review provides an appropriate assessment of the current state of this field. This manuscript is constructed well and the authors took a logical approach to present the different aspects of oral vaccine design as they move from topic to topic providing a brief overview of each, while providing the pertinent references. In addition, the expert opinion provides a clear summary of the field and direction on considerations for future efforts.

One aspect of this field that would strengthen the manuscript if included is the aspect of sexually dimorphic effects of oral vaccination and the racial and ethnic differences observed with oral vaccines. Importantly, sex differences have been observed in children and adults in response to vaccination (Flanagan et al Annu Rev Cell Dev Biol 2017) including increased immunogenicity and adverse events. These observations have been made with the oral polio and oral typhoid vaccines, which are relevant here and might instruct how we consider vaccination for different cohorts.

In addition, racial and ethnic differences in oral vaccination have also been identified, although less well studied, but evidence of lower rates of vaccine efficacy due to enteropathy have been reported. (Naylor EBioMed 2015) Other conditions have also been implicated in the lack of vaccine efficacy in different populations (Levine BMC Biol 2010; Patriarca, Rev Infect Dis 1991; Triki Vaccine 1997). These data suggest either microbiota, social determinants of health, or both are critical considerations for oral vaccine design.
